# Emergence and diversification of a host-parasite RNA ecosystem through Darwinian evolution

**Taro Furubayashi[1], Kensuke Ueda[2], Yohsuke Bansho[3], Daisuke Motooka[4], Shota Nakamura[4], Ryo Mizuuchi[5,6], Norikazu Ichihashi[2,3,5,7]\***

[1]Laboratoire Gulliver, CNRS, ESPCI Paris, PSL Research University, Paris, France; [2]Department of Life Science, Graduate School of Arts and Science, The University of Tokyo, Tokyo, Japan; [3]Graduate School of Frontier Biosciences, Osaka University, Osaka, Japan; [4]Research Institute for Microbial Diseases, Osaka University, Osaka, Japan; [5]Komaba Institute for Science, The University of Tokyo, Tokyo, Japan; [6]JST, PRESTO, Kawaguchi, Japan; [7]Universal Biology Institute, The University of Tokyo, Tokyo, Japan

**Abstract** In prebiotic evolution, molecular self-replicators are considered to develop into diverse, complex living organisms. The appearance of parasitic replicators is believed inevitable in this process. However, the role of parasitic replicators in prebiotic evolution remains elusive. Here, we demonstrated experimental coevolution of RNA self-replicators (host RNAs) and emerging parasitic replicators (parasitic RNAs) using an RNA-protein replication system we developed. During a long-term replication experiment, a clonal population of the host RNA turned into an evolving host-parasite ecosystem through the continuous emergence of new types of host and parasitic RNAs produced by replication errors. The host and parasitic RNAs diversified into at least two and three different lineages, respectively, and they exhibited evolutionary arms-race dynamics. The parasitic RNA accumulated unique mutations, thus adding a new genetic variation to the whole replicator ensemble. These results provide the first experimental evidence that the coevolutionary interplay between host-parasite molecules plays a key role in generating diversity and complexity in prebiotic molecular evolution.

**\*For correspondence:**
ichihashi@bio.c.u-tokyo.ac.jp

**Competing interests:** The authors declare that no competing interests exist.

## Introduction

Host-parasite coevolution is at the center of the entire course of biological evolution (*Claverie, 2006*; *Forterre and Prangishvili, 2009*; *Koonin and Dolja, 2013*; *Koskella and Brockhurst, 2014*). Parasitic replicators, such as viruses, are the most prosperous biological entities (*Bergh et al., 1989*; *Suttle, 2007*) that offer ever-changing selection pressure and genetic reservoirs in the global biosphere. The development of the sophisticated adaptive immunity (*Müller et al., 2018*) that prevails in all domains of life is a hallmark of the power of host-parasite coevolution, and accumulating evidence highlights the potential key roles of parasites in the development of the basic biological architectures and functions (*Claverie, 2006*; *Deininger et al., 2003*; *Elbarbary et al., 2016*; *Forterre, 2013*; *Forterre and Prangishvili, 2009*; *Iranzo et al., 2014*; *Koonin and Dolja, 2013*; *Koskella and Brockhurst, 2014*).

Parasitic replicators have probably worked as evolutionary drivers since the prebiological era of molecular replication (*Higgs and Lehman, 2015*; *Joyce and Szostak, 2018*; *Orgel, 1992*; *Szathmáry and Maynard Smith, 1997*; *Wochner et al., 2011*). Even in a simplest form of replication systems, parasites inevitably appear through a functional loss of self-replicating molecules and threaten the sustainability of the replication system (*Bansho et al., 2012*; *Koonin et al., 2017*).

Theoretical studies suggested that spatial structures, such as cell-like compartments, allow self-replicators (i.e. hosts) to survive by limiting the propagation of parasitic replicators (*Bresch et al., 1980*; *Furubayashi and Ichihashi, 2018*; *Szathmáry and Demeter, 1987*; *Takeuchi and Hogeweg, 2009*). Subsequent experimental studies demonstrated that the compartmentalization strategy effectively support the replication of host replicators in the presence of parasitic replicators (*Bansho et al., 2016*; *Bansho et al., 2012*; *Matsumura et al., 2016*; *Mizuuchi and Ichihashi, 2018*).

In a previous study (*Ichihashi et al., 2013*), we constructed an RNA replication system consisting of an artificial genomic RNA and a reconstituted translation system of *Escherichia coli* (*Shimizu et al., 2001*) encapsulated in water-in-oil droplets, to study how a simple molecular system develops through Darwinian evolution. In this system, the artificial genomic RNA (host RNA) replicates through the translation of the self-encoded replicase subunit. During replication, a deletion mutant of the host RNA (parasitic RNA), which lost the encoded replicase subunit gene, spontaneously appears and replicates by freeriding the replicase provided by the host RNA. Through serial nutrient supply and dilution, the host and parasitic RNAs in water-in-oil droplets undergo repeated error-prone replication and natural selection processes, that is Darwinian evolution.

In a subsequent study (*Bansho et al., 2016*), we performed a serial transfer replication experiment of the aforementioned RNA replication system to study the evolutionary process of the host and parasitic RNA replicators. We reported that the host and parasitic RNAs showed oscillating population dynamics and that the host RNA acquired a certain level of parasite-resistance in the final rounds of the replication experiment (43 rounds, 215 hr). However, we did not observe counter-adaptive evolution of the parasitic RNA to the host RNA, and the coevolutionary process of the host and parasitic RNAs remains unclear.

In this study, we reasoned that a much longer time may be necessary for coevolution of the host and parasitic RNA replicators; hence, we extended the replication experiment by an additional 77 rounds (385 hr). To understand their evolutionary dynamics during the replication experiment, we performed sequence analysis of the host and parasitic RNAs. We also conducted competitive replication assays using evolved host and parasitic RNA clones to confirm the coevolution of the host and parasitic RNAs. Moreover, we fully reanalyzed the host-parasite RNA population (up to 43 rounds) partially reported earlier (*Bansho et al., 2016*). In this paper, we present an analysis of 120 rounds (600 hr) of a longer term replication experiment, incorporating new data.

## Results

### RNA replication system

The RNA replication system used in this study consists of two classes of single-stranded RNAs (host and parasitic RNAs) and a reconstituted translation system of *E. coli* (*Shimizu et al., 2001*; *Figure 1A*). A distinctive feature of the host and parasitic RNAs is the capability of providing an RNA replicase (Qβ replicase). The host RNA provides the catalytic β-subunit of the replicase (via translation), which forms active replicase by associating with EF-Tu and EF-Ts subunits in the translation system, whereas the parasitic RNA lacks the intact gene. The host RNA replicates using the self-provided replicase, whereas the parasitic RNA relies on the host-provided replicase. We used a clone from round 128 in our previous study (*Ichihashi et al., 2013*) as the original host RNA because it replicates fast and had been characterized in detail.

In this system, parasitic RNAs spontaneously emerge from the host RNA by deleting the internal replicase gene plausibly through nonhomologous recombination (*Bansho et al., 2012*; *Chetverin et al., 1997*). The parasitic RNAs reported previously have similar sizes (~200 nt). We refer to parasitic RNA of this size as 'parasite-α'. Parasite-α replicates much faster than the original host RNA (~2040 nt), owing to its smaller size, and thus inhibits the host replication through competition for the replicase. The replication with Qβ replicase is error-prone, approximately $1.0 \times 10^{-5}$ per base (*García-Villada and Drake, 2012*), and mutations are randomly introduced into the host and parasitic RNAs during the replication reaction.

### Long-term replication experiment

We performed a long-term replication experiment of the host and parasitic RNAs. The replication reaction was performed in a water-in-oil emulsion (~$2 \times 10^9$ droplets in each round) by repeating a

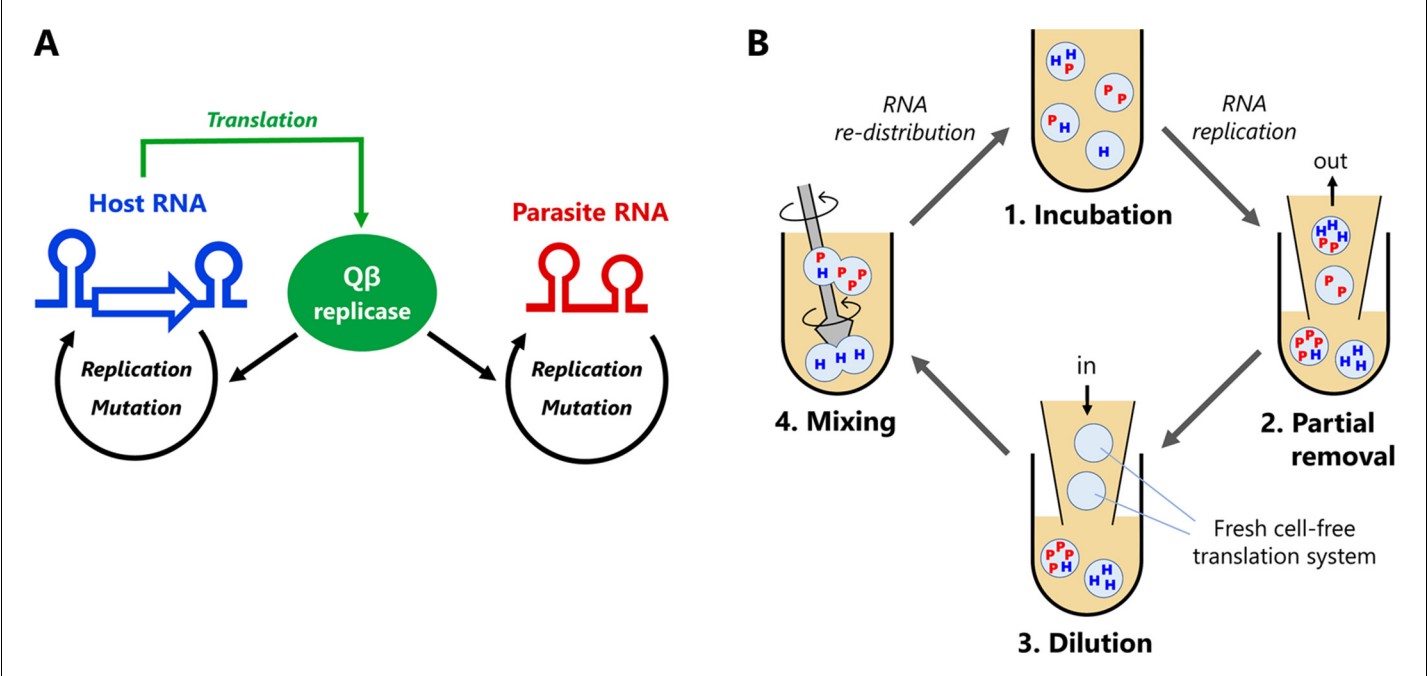

**Figure 1.** Host and parasitic RNA replication system. (**A**) Replication scheme of the host and parasitic RNAs. The host RNA encodes the Qβ replicase subunit, whereas the parasitic RNA does not. Both RNAs are replicated by the translated Qβ replicase in the reconstituted translation system of *E. coli*. (**B**) Replication-dilution cycle for a long-term replication experiment. The host RNA is encapsulated in water-in-oil droplets with ~ 2 μm diameter. The parasitic RNA spontaneously appears. (1) The droplets are incubated at 37°C for 5 hr for translation and replication. (2) Eighty percent of the droplets are removed and (3) diluted with new droplets containing the translation system (i.e. five-fold dilution). (4) Diluted droplets are vigorously mixed to induce fusion and division among the droplets. We repeated this cycle for 120 rounds. The reaction volume was 1 mL, with 1% aqueous phase, corresponding to ~ $10^8$ droplets.

fusion-division cycle with the supply of new droplets containing the translation system (*Figure 1B*). A single round of the experiment consisted of four steps: 1) incubation, 2) partial removal, 3) dilution, and 4) mixing. In the incubation process, the water-in-oil droplets were incubated at 37°C for 5 hr to induce internal translation and RNA replication reactions. We started with a clonal population of the host RNA (1 nM, ~6 × $10^9$ molecules) without parasite-α, which was, however, detected within two rounds. In the partial removal process, we removed 80% of the water-in-oil droplets. In the dilution process, we substituted them with new water-in-oil droplets containing the cell-free translation system (i.e. five-fold dilution). In the mixing process, droplets were vigorously mixed with a homogenizer to induce fusion and division among the droplets and allow the mixing of RNAs and other components. This replication-dilution cycle does not require manual mutagenesis, selection procedures, and control of the RNA copy number in the droplets, allowing easy implementation of long-term in vitro molecular evolution. We repeated this cycle for 120 rounds (600 hr) in total. All the following results were derived from this single long-term replication experiment.

## Population dynamics of host and parasitic RNAs

We measured the concentrations of the host and parasitic RNAs after every incubation process (*Figure 2A*). The host RNA was measured using quantitative PCR after reverse transcription (RT-qPCR). The parasitic RNA was measured using the band intensity after polyacrylamide gel electrophoresis (*Figure 2—figure supplement 1*) because these parasites were deletion mutants of the host RNA and could not be uniquely targeted by RT-qPCR. In some rounds (7–12, 16–22, and 75–84), the parasitic RNAs were under the detection limit (less than ~30 nM) and not visible due to the lower sensitivity of gel analysis compared to that of RT-qPCR.

The population dynamics of the host and parasitic RNAs gradually changed throughout the rounds (*Figure 2A*). In the early stage (rounds one to ~35), the host RNA and parasite-α showed a relatively regular oscillation pattern caused by competition between the host and parasitic RNAs in

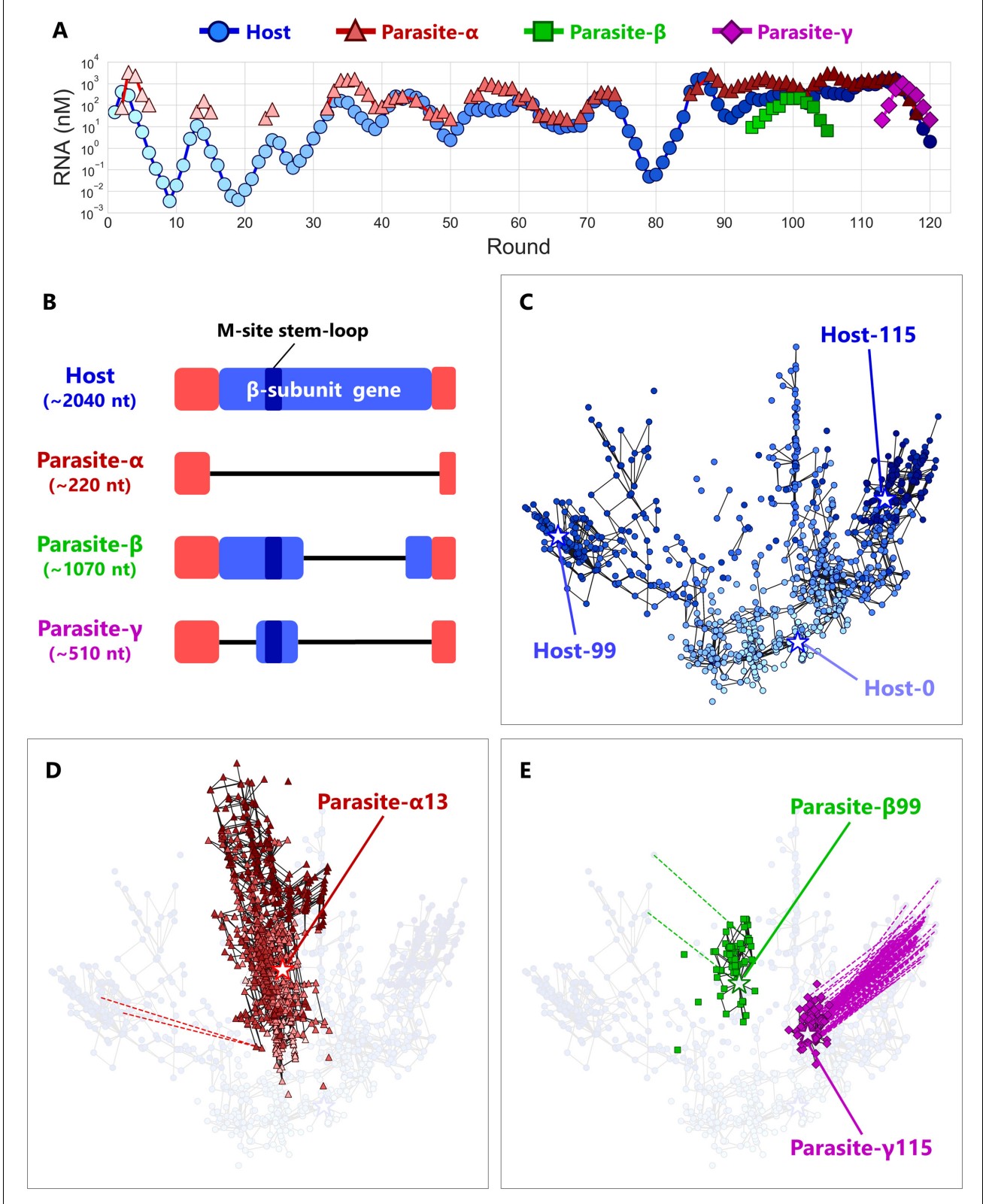

**Figure 2.** Coevolutionary dynamics of host and parasitic RNAs. (**A**) Population dynamics of the host and parasitic RNAs during a long-term replication experiment. In the regions without points, parasitic RNA concentrations were under the detection limits (<30 nM) of the gel analysis. Three different parasitic species (α, β, and γ) are classified based on their sizes. (**B**) Schematic representation of the sequence alignments of the host and parasitic RNA species. The terminal regions (red) of all the RNA species are derived by the replicase MDV-1 (*Mills et al., 1973*), from a small replicable RNA. The β-

Figure 2 continued

subunit encoding regions are shown in blue, and the branched stem-loop of the M-site, one of the binding sites for Qβ replicase, is also indicated. Deleted regions are shown using black lines. (**C, D, E**) 2D maps of the dominant RNA genotypes for the host RNA (**C**), parasite-α (**D**), and parasite-β and parasite-γ (**E**). The top 90 dominant genotypes were plotted for each round. A point represents each genotype. The color depths are consistent with those in (**A**). Black lines connect pairs of points one Hamming distance apart in the same RNA species. A broken line connects a pair of points zero Hamming distance apart (perfect match) in the different RNA species, ignoring the large deletion between host and parasitic RNAs. Stars represent the genotypes of the evolved RNAs used for the competitive replication assay shown in *Figure 4A*. The original host RNA is Host-0. Round-by-round data are shown in *Figure 3*.

The online version of this article includes the following source data and figure supplement(s) for figure 2:

**Source data 1.** Read numbers of deep sequencing.
**Source data 2.** Sequence data file after the alignment with the original host sequence, used to identify the 74 dominant mutations.
**Figure supplement 1.** The native polyacrylamide gel electrophoresis of the RNA mixture during the long-term replication experiment.
**Figure supplement 2.** Replication of Parasite-β99 and Parasite-γ115 without host species.
**Figure supplement 3.** Dominant mutations and fixation dynamics among host and parasitic RNAs.
**Figure supplement 4.** Phylogenic analysis of the host and parasite RNAs.
**Figure supplement 4—source data 1.** Alignment data used for *Figure 2—figure supplement 4*.

compartments. In this regime, the concentrations of parasite-α were higher than those of the host RNA. In the middle stage (rounds ~ 35 to~75), the concentrations of the host RNA increased, and the oscillation pattern became irregular. The elevation of the host RNA concentration can be attributed to less replication inhibition by parasite-α. We have previously reported that some nonsynonymous mutations in Qβ replicase encoded by the host RNA in round 43 selectively reduce the replication efficiency of parasite-α (*Bansho et al., 2016*). The prevalence of these mutations probably allows the host RNA population to maintain higher concentrations than that in the early stage. In the later stage (rounds ~ 95 to~116), the concentrations of the host and parasite-α further increased, and the oscillation pattern became more unclear. In this regime, we observed the appearance of new parasitic RNA species of different sizes and classified them as parasite-β (~1000 nt, green squares) and parasite-γ (~500 nt, purple diamonds) according to their sizes. We termed these new RNAs 'parasites' because each clone of these RNAs did not replicate alone (*Figure 2—figure supplement 2*). Such continuously changing population dynamics can be caused by successive adaptation processes between host and parasitic RNAs.

## Sequence analysis

To investigate the evolutionary dynamics of host-parasite RNA populations at the sequence level, we recovered RNA mixtures from 17 points (rounds 13, 24, 33, 39, 43, 50, 53, 60, 65, 72, 86, 91, 94, 99, 104, 110, and 115), and subjected them to reverse transcription followed by deep sequencing with PacBio RS II for the host, parasite-β, and parasite-γ and MiSeq for parasite-α. With PacBio RS II sequencing, we obtained 365–4143 reads for each class of RNA in the sequenced rounds. With MiSeq sequencing, we obtained ~5000 reads for each round (*Figure 2—source data 1*).

Sequence analysis revealed that four major RNA classes with different sizes existed in the long-term replication experiment, consistent with the band positions observed in the polyacrylamide gels:~2040 nt (the host),~220 nt (parasite-α),~1070 nt (parasite-β), and ~510 nt (parasite-γ). The sequences of all the classes of parasitic RNA shared a high degree of similarity with those of the host RNAs but lacked a large part of the replicase subunit gene (*Figure 2B*). The parasite-α sequence class lacks the entire gene. The parasite-β sequence class lacks approximately the 3' half of the gene, and parasite-γ sequence class further lacks ~25% of the remaining 5' region of the gene. Both parasite-β and parasite-γ retain a part of the M-site sequence, one of the recognition sites for Qβ replicase (*Meyer et al., 1981*; *Schuppli et al., 1998*), in the middle of the gene.

We then determined the dominant genotypes of all the classes of RNA (host, parasite-α, parasite-β, and parasite-γ). Although the RNA replication by Qβ replicase is error-prone and introduces many random mutations that produced quasi-species for each genotype, we focused on the consensus sequences that consist of mutations commonly found in the RNA population. We first identified 74 dominant mutations that were present in more than 10% of the population of each class of RNA in a sequenced round. The dominant mutations consisted of 60 base substitutions, four insertions, and 10 deletions in total (*Figure 2—figure supplement 3*). Then, we measured the frequencies of all the

genotypes composed of the combination of these 74 dominant mutations in every sequenced round for each class of RNA. All the genotypes and their frequencies are shown in the *Supplementary file 1*.

We then investigated the relationships of the detected genotypes. To visualize evolutionary trajectories, we calculated Hamming distances between all combinations of the top 90 genotypes of all the classes of RNA species in the sequenced rounds and then plotted them in a single two-dimensional (2D) map, using Principal Coordinate Analysis. RNA species-wise data are shown in *Figure 2C–E*, and round-wise data of all the RNA species are plotted together in *Figure 3* to recapitulate the evolutionary dynamics of the entire RNA population throughout the replication experiment (animation of these snapshots is provided in *Figure 3—animation 1*). A point represents each genotype, and the colors of points represent the rounds they appeared consistent with the colors of the markers in *Figure 2A*. A black line connects a pair of genotypes one Hamming distance apart in the same RNA class. We assigned zero distance for the large deletions between the host and parasitic RNAs. The host RNA genotypes gradually became distant from the original genotype (Host-0) as the rounds proceeded (*Figure 2C* and *Figure 3*). From round 0 to round 43, sequences diversified around the original genotype. Then, until round 72, most of the genotypes moved toward the upper-right branches. However, in round 86, a certain fraction of the genotypes shifted to the left branch and dominated until round 99. In round 104, most of the genotypes moved back to the right branch again and stayed there until round 115. These frequent changes in dominant lineages imply that the fittest genotype changes frequently during the long-term replication experiment.

The population of parasite-α represented a cluster distinct from host RNA populations (*Figure 2D*), and most of the genotypes were connected (i.e. one Hamming distance apart).

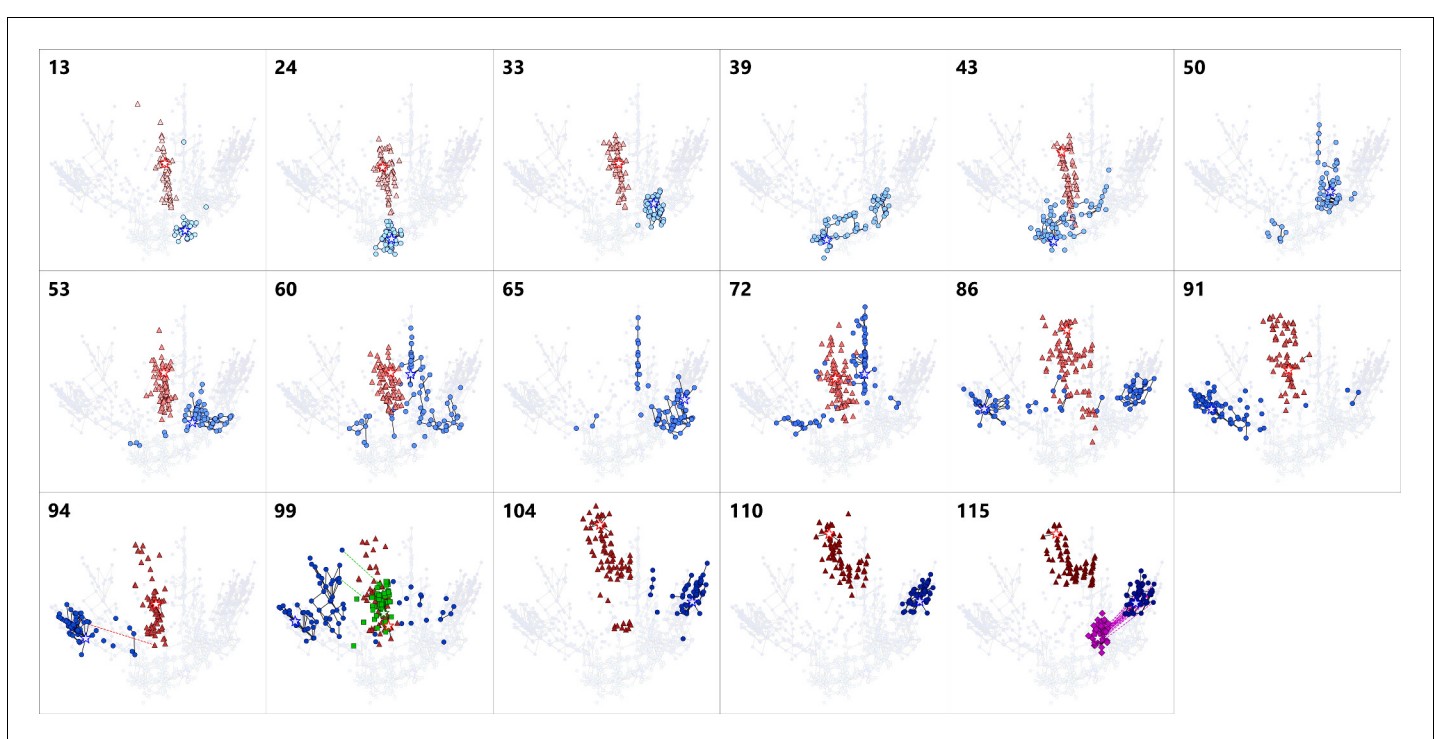

**Figure 3.** Series of snapshots of dominant RNA genotypes on 2D maps for the host RNA, parasite-α, -β and -γ. The upper-left numbers indicate the round. The top 90 dominant genotypes of each RNA species were plotted for each round. A point represents each genotype. The colors of points are consistent with *Figure 2*, the host (blue), parasite-α (red), -β (green), and -γ (purple). A star in each figure represents the most frequent genotype of the host (blue), parasite-α (red), -β (green), and -γ (purple). Black lines connect pairs of points one Hamming distance apart in the same RNA species. A broken line connects a pair of points zero Hamming distance apart in the different RNA species, ignoring the large deletion, which represent a plausible generation route of each parasite. Colors of the broken lines correspond to the host and parasite-α (red), the host and parasite-β (green), and the host and parasite-γ (purple). Parasite-α is not shown in the round-39, 50, and 65 because they could not have been recovered and sequenced.

**Figure 3—animation 1.** Animation of *Figure 3*.

https://elifesciences.org/articles/56038#fig3video1

Parasite-α did not show clear directionality throughout the long-term replication experiment (*Figure 3*). Interestingly, we identified 18 unique mutations specific to parasite-α (*Figure 2—figure supplement 3*), which were not found in the corresponding region of the host sequence. The persistence of the unique mutations of parasite-α and the differences in the mutational patterns of parasite-α and the coevolving hosts indicate that many of the new parasite-α genotypes were not newly generated from evolving host RNAs, and that parasite-α maintained its own lineage and evolved independently of the host RNA. We also observed the appearance of new parasite-α species from the evolved host RNAs owing to deletion. For example, a parasite-α genotype that appeared in round 94 perfectly matched with a host genotype in round 94 (connected with a red broken line in *Figures 2D* and *3*), except for a large internal deletion, suggesting that this parasite was generated from the evolving host through a deletion event. Note that we could not obtain the sequence data of parasite-α in rounds 39, 50, and 65 because the cDNA could not be recovered.

The populations of parasite-β and parasite-γ formed distinct clusters (*Figure 2E*), and most of the genotypes were closely related within each class (connected with one Hamming distance lines). Sequences of some parasite-β and parasite-γ perfectly matched with some dominant host RNAs coexisting in the same rounds as those connected with green or purple broken lines each, suggesting that these parasitic RNAs originated from the host RNAs. Unlike parasite-α, we found only 2 and 1 unique mutations for parasite-β and parasite-γ, respectively (*Figure 2—figure supplement 3*).

To understand the relationship between the host and parasite lineages, we performed phylogenic analysis of the top three most frequent genotypes of the host and parasite RNAs in all the sequenced rounds (*Figure 2—figure supplement 4*). The phylogenic tree contains two large branches: branch P (colored in red) contains most parasite-α and branch H (colored in blue) contains all the other RNAs. This result confirmed that parasite-α evolved independently. Branch H further contains two sub-branches: branch H1 contains all parasite-β and host RNAs in rounds 60–99, and branch H2 contains all parasite-γ and host RNAs during the early (until 65–86) and later (104-115) rounds. This result support that there are two lineages in the host RNAs, corresponding to the population rounds, Host-99 and Host-115, as shown in *Figure 2C*, and that parasite-β and parasite-γ are their respective descendants (*Figure 2E*). We could not find a clear trend in transition for parasite-α (i.e. in branch P). The earliest parasite-α in round 13 (indicated with red asterisks) were already distributed into different sub-branches, and those at later rounds existed within or around the sub-branches. This result indicates that many of the mutations that characterized parasite-α appeared and were fixed by round 13, and then parasite-α wandered around in the sequence space. It is also notable that a parasite-α genotype (Alpha 094R Rank2 with a green tick) are located within host clusters, indicating that it emerged from the evolved host in a later round.

## Competitive replication assay of host and parasitic RNAs

The diversification of host genotypes and the appearance of novel parasite classes can be a consequence of the coevolution between the hosts and parasites to adapt to each other. To test this possibility, we performed a series of competitive replication assays using three representative host and parasitic RNAs. We chose the most dominant host genotypes in rounds 0, 99, and 115 (Host-0, Host-99, and Host-115, respectively). For parasite-α, parasite-β, and parasite-γ, we chose the most dominant genotypes in rounds 13, 99, and 115 (Parasite-α13, Parasite-β99, and Parasite-γ115), respectively (sequences are available in *Supplementary file 1*). We mixed a pair of these host and parasitic RNA clones, according to their order of appearance, at an equivalent molarity, and performed competitive replication. The concentrations of replicated RNAs were measured by sequence-specific RT-qPCR (*Figure 4A*). RT-qPCR of parasites was possible in this experiment because we designed primers very specific to each parasite clone, which was not possible for the evolving RNA mixture containing various mutations. In the first pair (Host-0 vs Parasite-α13), Host-0 hardly replicated (less than 2-fold) and Parasite-α13 predominantly replicated (~200 fold), indicating that Parasite-α13 severely inhibits the original host replication, whereas in the second pair (Host-99 vs Parasite-α13), Host-99 efficiently replicated (~700 fold) with negligible replication of Parasite-α13, indicating that Host-99 acquired resistance to Parasite-α13. In the third pair (Host-99 vs Parasite-β99), Host-99 replicated efficiently (~1000 fold), but Parasite-β99 also replicated up to ~20 fold, indicating that Parasite-β99 acquired the ability to co-replicate with Host-99. In the fourth pair (Host-115 vs Parasite-β99), Host-115 repressed the replication of Parasite-β99 to less than twofold, indicating that Host-115 acquired the ability to evade co-replication of Parasite-β99. In the final pair (Host-115

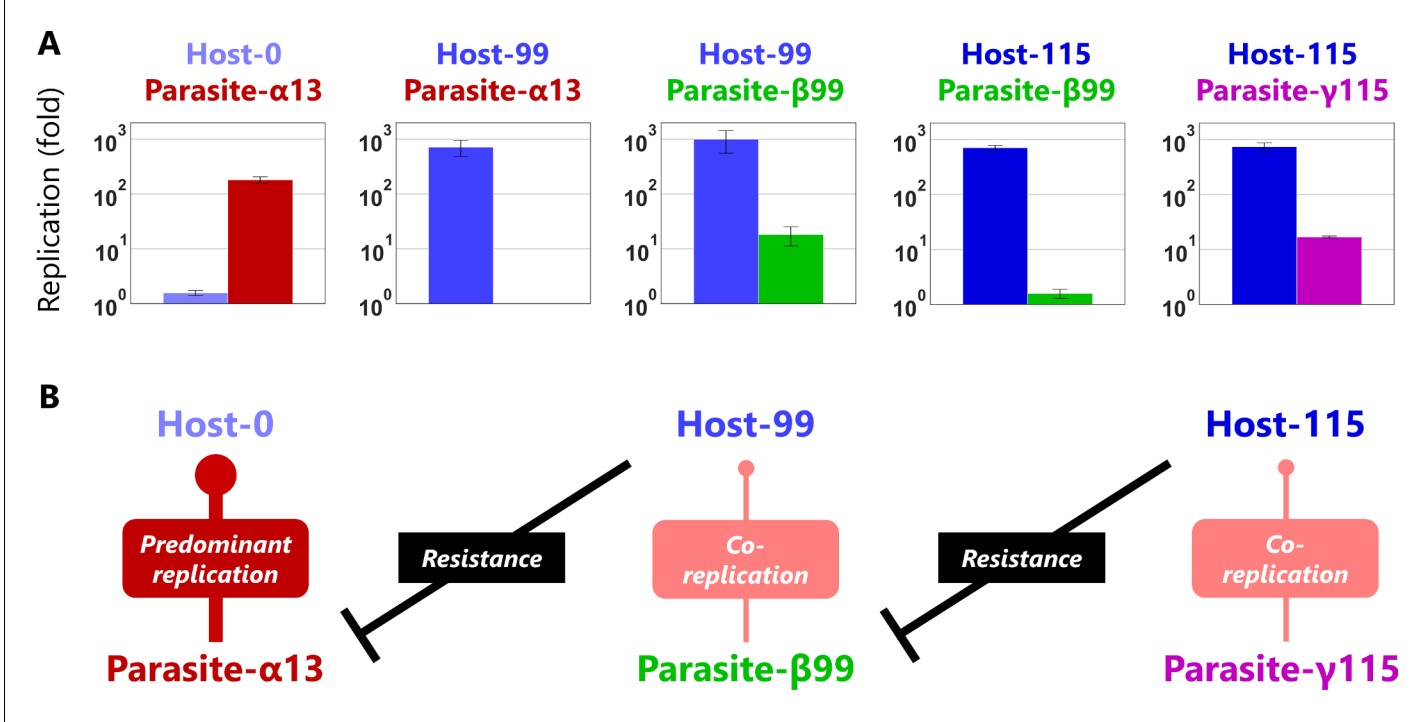

**Figure 4.** Evolutionary arms races between host and parasitic RNAs. (**A**) Competitive replication assays of each pair of the evolved host and parasitic RNA clones. The RNA replication reactions were performed with 10 nM of the host and parasitic RNAs for 3 hr, and each concentration was measured by sequence-specific RT-qPCR. Error bars represent standard errors of three independent competition assays. (**B**) Schematic representation of the host-parasite relationships among the RNA clones.

The online version of this article includes the following source data and figure supplement(s) for figure 4:

**Source data 1.** Dominant mutations in Host-99 and Host-115.

**Figure supplement 1.** Combinatorial competitive replication assay of Hosts-99 and −115 with Parasites-β99 and -γ115.

vs Parasite-γ115), Parasite-γ115 acquired the ability to replicate up to ~20 fold with Host-115. These results demonstrated that successive counter-adaptive evolution (i.e. evolutionary arms races) occurred among the host and parasitic RNAs, as schematically illustrated in *Figure 4B*. We also examined the Host-99 vs Parasite-γ115 relationship and found that Parasite-γ115 was hardly replicated by Host-99 (*Figure 4—figure supplement 1*), indicating that parasite-β and parasite-γ are specifically parasitic to Host-99 and Host-115, respectively.

## Discussion

Coevolution of host and parasitic replicators is a major driver in the evolution of life. In this study, we investigated the Darwinian evolution process of an RNA replication system and demonstrated the emergence of a host-parasite ecosystem in which new types of host and parasitic RNAs appeared successively and exhibited antagonistic coevolutionary dynamics. Notably, all the host and parasite RNAs that appeared in the long-term replication experiment are descendants of the single host RNA. Throughout the replication experiment, the host RNA continued to evolve and diverge into distinct evolutionary branches in a sequence space (*Figures 2C* and *3*, and *Figure 2—figure supplement 4*), which stands in sharp contrast to the previously reported unidirectional and rapidly slowing evolution of the host RNA in the absence of frequent interactions with parasitic RNAs (*Ichihashi et al., 2013*). The diversification of parasitic RNAs into three distinct parasite classes is also a new phenomenon that was not observed in our previous study (*Bansho et al., 2016*). The dynamic change of the host-parasite genotypes (*Figures 2C–E* and *3*) and phenotypes (*Figure 4* and *Figure 4—figure supplement 1*) indicates that evolving parasites could have driven the diversification of the host RNA by providing varying selection pressure. In fact, the diverged host RNAs (Host-99 and Host-115) had very different mutational patterns, with only a few shared mutations

(*Figure 4—source data 1*), supporting the possibility that parasites with different phenotypes promoted the evolution of different strategies of host RNAs, as discussed below. This coevolution-driven diversification is consistent with the consequence of natural host-parasite coevolution (*Bohannan and Lenski, 2000*; *Buckling and Rainey, 2002*; *Ebert, 2008*; *Thompson, 1999*; *Woolhouse et al., 2002*) and simulated in silico evolution (*Takeuchi and Hogeweg, 2008*; *Zaman et al., 2011*). Therefore, evolutionary arms races between host-parasite molecules may have been an important mechanism to generate and maintain diversity in molecular ecosystems before the origin of life.

High-resolution sequence tracking of RNA populations allowed the observation of reciprocal host-parasite mutational dynamics underlying evolutionary arms-race history. In the first sequenced round (round 13), parasite-$\alpha$ had already accumulated as many as seven mutations, whereas the host did not have fixed beneficial mutations (*Figure 2—figure supplement 3*). Six out of these seven mutations are unique mutations of parasite-$\alpha$ and five of them continued to exist until the final round (round 115), implying that parasite-$\alpha$, which appeared in the early stage of evolution, maintained continuous lineage and persisted throughout the long-term replication experiment. A clear sign of adaptive evolution of the host first appeared in round 39, consistent with the elevation of host RNA concentration (*Figure 2A*). Nonsynonymous mutations (Lys208Asp, Leu448Arg, and Gln459Arg) in Q$\beta$ replicase that occurred in this round were found responsible for the resistance against parasite-$\alpha$ in our previous study (*Bansho et al., 2016*). Interestingly, between rounds 50 and 72, many mutations appeared and disappeared in both the host and parasitic RNAs (*Figure 2—figure supplement 3*), and both genotypes wandered around the sequence space (*Figure 3*), suggesting that a co-evolutionary event had occurred. For example, the parasite-$\alpha$ RNA concentration suddenly recovered in round 53 (*Figure 2A*), and quick accumulation of the C1986U mutation might have been beneficial. Thereafter, the host accumulated two non-synonymous mutations (A452G and A626G), and the C1986U mutation disappeared from parasite-$\alpha$ in round 65. From round 86, the host accumulated as many as 11 mutations simultaneously, and the population rapidly converged toward the left branch, including Host-99, in the sequence space (*Figures 2C* and *3*). In round 104, coincident with the rise of parasite-$\beta$, all the 11 mutations that characterizes the left-branch hosts almost disappeared from the population. Instead, eight new mutations accumulated in the host, and the population quickly moved toward the right branch, including Host-115. The genotype of parasite-$\alpha$ also drastically changed in round 104, suggesting its adaptive evolution to the hosts in the right branch. Upon the invasion of parasite-$\gamma$ in round 115, some mutations (e.g. C72U, C259U, U501C, and A851G) appeared and disappeared in the host population. These host-parasite mutational dynamics exhibit how coevolution progressed during the replication experiment. Finally, we mention that we searched for possible recombination events in the host and parasite sequences throughout the replication experiment, using the RDP4 program (*Martin et al., 2015*), but did not detect a recombination signal.

The mutational patterns of the host and parasitic RNAs in this study suggest an interesting possibility that parasites could bring about new information in a molecular population. Parasite-$\alpha$ accumulated nine dominant mutations in the 3'-UTR, whereas the host RNA never accumulated dominant mutations during long-term evolution in the region (*Figure 2—figure supplement 3*). This result suggests that mutations in the 3'-UTR of the host RNA are severely limited (constraint imposed by translation efficiency). Evolving and persistent parasitic molecules with less mutational constraints may add genetic novelties to the whole molecular ensemble and play a role in the evolution of complexity (*Adami et al., 2000*).

It is generally believed that evolution progresses toward more complexity in nature (*Petrov, 2001*; *Sharov, 2006*); however, genome reduction is also a popular mode of evolution (*Albalat and Cañestro, 2016*; *Morris et al., 2012*; *Wolf and Koonin, 2013*). Therefore, the condition in which genomic information expands and reduces is a fundamental question. Especially in the prebiotic molecular evolution context, the benefit of genome reduction is obvious because shorter molecules can replicate faster. In fact, in previous in vitro Darwinian evolution experiments (*Ichihashi et al., 2013*; *Mills et al., 1967*), evolution favored shorter genomes for faster replication; selection for longer genomes has not been reported. A remarkable phenomenon in our study is that longer parasites with a long RNA genome appeared after long-term evolution (after 94 rounds). The new parasites, parasite-$\beta$ and parasite-$\gamma$, became longer because they retained a part of the M-site sequence, a recognition site for Q$\beta$ replicase (*Meyer et al., 1981*; *Schuppli et al., 1998*), which did not exist in

parasite-α. A plausible scenario for the appearance of these parasites is as follows: (1) parasite-α first invaded the system, taking advantage of its short genome for faster replication; (2) the host RNA evolved the specificity of Qβ replicase to host-specific sequences (including the M-site) to circumvent parasite-α; and (3) the new parasites invaded the system because they retained evolved M-sites that were recognized by evolved Qβ replicases when they appeared from the evolved hosts. According to this scenario, the new parasites appeared to be expanding the genomic information to cope with the evolved strategy of the host RNA, which may be consistent with recent theoretical studies that suggest that host-parasite antagonistic coevolution is an effective mechanism to increase the complexity of individuals (*Seoane and Solé, 2019*; *Zaman et al., 2014*). The next important question would be whether further long-term coevolution can lead to genome expansion of the host RNA.

A typical phenomenon in host-parasite coevolution is Red Queen dynamics (*Rabajante et al., 2015*; *Van Valen, 1973*), in which host and parasite populations oscillate due to persistent replacement of dominant hosts and parasites. The host-parasite RNA population in our replication experiment exhibited Red Queen dynamics with a remarkable feature of damping fluctuations. One possible reason for the damped oscillation is simply the elevation of the average parasite resistance against parasite-α in the evolved host RNA population, which may be partly supported by the weakened inhibition of the host replication by the parasitic RNAs in later rounds (*Figure 4*). Another possibility is that increased diversity (*Figures 2C–E* and *3*) allows competition among various types of host and parasitic RNAs to average the population dynamics. A study on *Daphnia* and its parasite also reported damped long-term host-parasite Red Queen coevolutionary dynamics and suggested that the increased host diversity as a consequence of coevolution could decrease fluctuations in host-parasite Red Queen dynamics (*Decaestecker et al., 2013*). Theoretical studies also suggest that intra-species phenotypic divergence (*Van der Laan and Hogeweg, 1995*) and mutation rate elevation (*Kaneko and Ikegami, 1992*) can lead to stable host-parasite (or prey-predator) coexistence with small-amplitude oscillation. Our simple and fast-evolving host-parasite RNA replication system may offer a useful platform to investigate these tendencies of ecological and evolutionary dynamics of hosts and parasites and further pursue an exciting evolution scenario, such as the emergence of cooperation between host-parasite replicators.

## Materials and methods

### Long-term replication experiment

In this study, we performed an additional 77 rounds of replication using the RNA population of round 43 of a previous experiment, using the same method (*Bansho et al., 2016*). In this method, initially, 10 μL of the reconstituted *E. coli* translation system (*Shimizu et al., 2001*) containing 1 nM of the original host RNA, Host-0, and the round 128 clone in a previous study (*Ichihashi et al., 2013*), was mixed with 1 mL of a buffer-saturated oil prepared as described previously (*Ichihashi et al., 2013*), using a homogenizer (POLYTRON PT-1300D; KINEMATICA), at 16,000 rpm for 1 min on ice. The water-in-oil droplets were incubated at 37°C for 5 hr to induce protein translation and RNA replication reactions. For the next round of RNA replication, a fraction of the water-in-oil droplets (200 μL) was transferred and mixed with the new buffer-saturated oil (800 μL) and translation system (10 μL), using the homogenizer, at 16,000 rpm for 1 min on ice, and then incubated at 37°C for 5 hr. The average diameter of the water-in-oil droplets was ~2 μm (*Bansho et al., 2016*), and the number of droplets was ~$2 \times 10^9$. After the incubation step in each round, RNA concentrations were measured as described below. The composition of the translation system has been described previously (*Bansho et al., 2016*).

### Measurement of host RNA concentrations

After the incubation step, the water-in-oil droplets were diluted 10,000-fold with 1 mM EDTA (pH 8.0) and subjected to RT-qPCR (PrimeScript One Step RT-PCR Kit (TaKaRa)) with primers 1 and 2 after heating at 95°C for 5 min. These primers specifically bind to the host RNA. To draw a standard curve in RT-qPCR, dilution series of the water-in-oil droplets containing the original host RNA diluted 10,000-fold with 1 mM EDTA were used.

## Measurement of parasitic RNA concentrations

To determine the concentrations of the parasitic RNAs that appeared during the long-term replication experiment (*Figure 2A*), polyacrylamide gel electrophoresis was performed, followed by quantification of the fluorescence intensities of the parasitic RNA bands using ImageJ. The water phases were collected from the water-in-oil droplets after the incubation step at each round, and RNAs were purified with spin columns (RNeasy, QIAGEN). The purified RNA samples and dilution series of the standard parasitic RNA (S222 RNA [*Hosoda et al., 2007*]) were subjected to 8% polyacrylamide gel electrophoresis with 0.1% SDS in TBE buffer (pH 8.4) containing tris(hydroxymethyl)aminomethane (100 mM), boric acid (90 mM), and EDTA (1 mM), followed by staining with SYBR Green II (Takara). The fluorescence intensities of the parasitic RNA bands were quantified, and the concentrations were determined based on the standard curve drawn with the dilution series of the standard parasitic RNA bands.

In a previous study (*Bansho et al., 2016*), we determined the parasitic RNA concentration from the replication kinetics using a purified Qβ replicase, and the detection limit was lower than that of this study. This method could not be employed in this study because it was unable to distinguish the different classes of parasitic RNAs that appeared.

## Sequence analysis

The RNA mixtures of rounds 13, 24, 33, 39, 43, 50, 53, 60, 65, 72, 86, 91, 94, 99, 104, 110, and 115 in the long-term replication experiment were purified with spin columns (RNeasy, QIAGEN). The purified RNAs were reverse-transcribed using PrimeScript reverse transcriptase (Takara) and primer three and then PCR-amplified using primers 3 and 4. The PCR products were subjected to agarose gel electrophoresis, and the bands corresponding to the host and parasitic cDNA were separately extracted using E-gel CloneWell (Thermo Fisher Scientific). The host, parasite-β, and parasite-γ were sequenced using PacBio RS II with C4/P6 chemistry (Pacific Biosciences), and parasite-α was sequenced using MiSeq (Illumina). To reduce read errors in the PacBio RS II sequencing, we used circular consensus sequencing (CCS) reads comprising at least five and ten reads for the host and parasites, respectively, to eliminate sequence errors. The read numbers in the *Supplementary file 1* indicates those of CCS. All the sequence reads were subjected to sequence alignment with a reference sequence (the original host sequence) for each molecular species (i.e. the host, parasite-α, parasite-β, and parasite-γ), using MAFFT v7.294b with the FFT-NS-2 algorithm (*Katoh et al., 2002*). The sequence data after alignment was provided as *Figure 2—source data 2*. Frequencies of mutations were calculated for each sample, and 74 dominant mutations that were present in more than 10% of the population of each class of RNA in a sequenced round were identified (*Figure 2—figure supplement 3*). These mutations were located in 72 sites (i.e. a few mutations were introduced in the same sites). In the subsequent analysis, we focused on only the genotypes associated with these 72 mutation sites. Focusing only on these dominant mutation sites minimizes the influence of remaining sequencing errors and non-dominant mutations in the other sites.

## Mapping dominant genotypes in two-dimensional space

Among the genotypes associated with the 72 mutation sites, the top 90 most dominant genotypes were identified for each host and parasitic species in each round. Hamming distances between all the pairs of genotypes were calculated, and a square distance matrix $D$, whose $i,j$-th component $d_{ij}$ represented the square of the Hamming distance between the $i$-th and $j$-th genotypes, was constructed. Using principal coordinate analysis on the square distance matrix $D$, the positions of each genotype were determined. Matrix $D$ was transformed into a kernel matrix $K = -1/2CDC$, where $C$ is the centering matrix. $\lambda_k$ and $e_k \equiv (e_{k1}, e_{k2}, \ldots, e_{kM})$ denote the $k$-th eigenvalue and the $k$-th normalized eigenvector, where $\lambda_1 > \lambda_2 > \ldots > \lambda_M$ and $|e_k| = 1$ for all $k$ and $M$ is the dimension of the kernel matrix $K$. The eigenvalues and eigenvectors of the kernel matrix $K$ were calculated, and the $i$-th genotype was plotted in two-dimensional space with a coordinate described as follows:

$$(X(i), y(i)) = (\sqrt{\lambda_1 e_{1i}}, -\sqrt{\lambda_2 e_{2i}})$$

## Phylogenic analysis of parasite RNA species by the maximum likelihood method

We extracted the top three most frequent sequences of the host, parasite-α, parasite-β, and parasite-γ from every sequenced round and conducted evolutionary analyses using MEGA X (*Kumar et al., 2018*). The evolutionary history was inferred using the maximum likelihood method and Tamura-Nei model (*Tamura and Nei, 1993*). Initial tree(s) for the heuristic search were obtained automatically by applying the Neighbor-Join and BioNJ algorithms to a matrix of pairwise distances estimated using the Tamura-Nei model and then selecting the topology with a superior log likelihood value. The gap/missing dataset treatment option was set as 'complete deletion'.

## Recombination scan of host and parasite RNAs using RDP4

We extracted the top 50 most frequent sequences of the host, parasite-α, parasite-β, and parasite-γ from every sequenced round and created a FASTA file. Using the RDP4 program (*Martin et al., 2015*), we performed a full exploratory recombination scan of the FASTA file with the RDP, Chimaera, GENECONV, 3Seq, and MaxChi algorithms.

## Competitive replication assay of host and parasitic RNAs

Six plasmids, each containing the T7 promoter and cDNA sequences of Host-0, Host-99, Host-115, Parasite-α13, Parasite-β99, and Parasite-γ115, were constructed using the gene synthesis service of Eurofins Genomics. Each RNA was synthesized from the plasmids digested with SmaI by in vitro transcription with T7 RNA polymerase (TaKaRa), in accordance with a previous study (*Yumura et al., 2017*). We mixed 10 nM each of host and parasitic RNAs in the cell-free translation system and incubated them at 37°C for 3 hr. The concentrations of the host and the parasitic RNAs were measured by RT-qPCR (PrimeScript One Step RT-PCR Kit (TaKaRa)) with sequence-specific primers (*Supplementary file 1*).

# Acknowledgements

We thank Nobuto Takeuchi, Kunihiko Kaneko, Yoshihiro Sakatani, Yannick Rondelez, and Tetsuya Yomo for the useful discussions and comments. This work was supported by JSPS KAKENHI grant numbers JP15KT0080, JP15H04407, and JP17J01023; the 'Innovation inspired by Nature' Research Support Program; SEKISUI CHEMICAL CO., LTD.; and the Astrobiology Center Program of the National Institutes of Natural Sciences (NINS) (Grant Number AB021005).

# Additional information

### Funding

| Funder | Grant reference number | Author |
|---|---|---|
| Japan Society for the Promotion of Science | JP15KT0080 | Norikazu Ichihashi |
| Japan Society for the Promotion of Science | JP15H04407 | Norikazu Ichihashi |
| Japan Society for the Promotion of Science | JP17J01023 | Taro Furubayashi |
| Sekisui Chemical | Innovations Inspired by Nature Research Support Program | Norikazu Ichihashi |
| National Institutes of Natural Sciences | Astrobiology Center Program AB021005 | Norikazu Ichihashi |

The funders had no role in study design, data collection and interpretation, or the decision to submit the work for publication.

## Author contributions
Taro Furubayashi, Conceptualization, Data curation, Formal analysis, Investigation, Visualization, Methodology, Writing - original draft; Kensuke Ueda, Investigation; Yohsuke Bansho, Data curation, Investigation, Methodology; Daisuke Motooka, Shota Nakamura, Resources, Data curation, Methodology; Ryo Mizuuchi, Investigation, Writing - review and editing; Norikazu Ichihashi, Conceptualization, Supervision, Funding acquisition, Visualization, Project administration, Writing - review and editing

## Author ORCIDs
Taro Furubayashi (iD) https://orcid.org/0000-0002-2549-5156
Norikazu Ichihashi (iD) https://orcid.org/0000-0001-7087-2718

## Decision letter and Author response
Decision letter https://doi.org/10.7554/eLife.56038.sa1
Author response https://doi.org/10.7554/eLife.56038.sa2

# Additional files

## Supplementary files
- Supplementary file 1. Clone sequence, primer list, the dominant genotypes and their frequencies.
- Transparent reporting form

## Data availability
All data generated or analyzed during this study are included in the manuscript and supporting files. Source data files have been provided for Figures 2 and 4.

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
