## [Decision Letter]

**Acceptance summary:**

During evolution, the emergence of parasites or cheaters is inevitable, but in the
end, co-evolution results in a stalemate between the parasite and its hosts. This
exciting paper uses the simplest possible system to investigate the emergence of a
parasite and subsequent co-evolution with the host: A self-replicating RNA. This
work and the elegance of this particular system set the stage for obtaining
mechanistic insights into how mutations increase fitness of the parasite, followed
by corresponding mutations that increase host fitness, followed by mutations that
increase parasite fitness, and so on in perpetuity.

**Decision letter after peer review:**

Thank you for submitting your article "Emergence and diversification of a
host-parasite RNA ecosystem through Darwinian evolution" for consideration by
*eLife*. Your article has been reviewed by three peer reviewers,
and the evaluation has been overseen by Detlef Weigel as the Senior and Reviewing
Editor. The following individuals involved in review of your submission have agreed
to reveal their identity: Eörs Szathmáry (Reviewer #1); Paulien Hogeweg (Reviewer
#2); Erik Hom (Reviewer #3).

The reviewers have discussed the reviews with one another and the Reviewing Editor
has drafted this decision to help you prepare a revised submission.

Summary:

This is an exciting and very relevant study about molecular in vitro co-evolution of
host and parasite RNA species. The generated data are interesting and provide a nice
avenue for understanding coevolutionary dynamics in an experimentally tractable way.
Although, as such experiments tend to do, they stop at the moment that tantalizing
new dynamics starts, the authors clearly demonstrate co-evolution of the RNA coding
for the replicase-β subunit (called host) and the immediately arising parasites
(lacking this) in a few competition experiments. This work (and indeed, the elegance
of this particular system) sets the stage for obtaining mechanistic insights into
how mutations lead to compensatory fitness increases in host and parasite along Red
Queen trajectories.

The agreement was that additional data analysis would enhance the work. Specific
suggestions are for a deeper analysis of lineage tracking and diversification
emergence (as suggested by the title), analysis of potential recombination events,
clarifications of methods, and broader contextualization of the findings with ideas
in the literature about host-parasite coevolution.

There are also a couple of suggestions for additional sequencing data, but we
recognize that it might currently be difficult to obtain such data. If that is the
case, please provide other explanations and/or adjust your claims.

Essential revisions:

1) The idea of an arms race in this system is attractive, but it raises questions
about the role of homologous copy-choice recombination in the host RNA population.
Under such conditions of directional selection (exerted by the parasites)
recombination should be advantageous. Is there a signal of this process in the
sequences of the host population?

2) It is confusing that different measurements of density for host (replicase) as
well as the different parasites are used. Please explain.

3) The population dynamics first shows strong oscillation, which later dampen till
t~75. Competition is only shown for parasite t-13 and host t=0 and parasite t=13 and
host t=99 (after one more deep oscillation). What happens in between? How can the
replicase sustain such high densities of host and parasite RNA, while that is
initially not the case?

4) Is the α-parasite after t=90 offspring of the older α parasite or a newly evolved
from the host by a large deletion?

5) The β and γ parasites are clearly offspring of the later hosts, each of one of the
evolved 'subspecies' of the host. The β parasites appear to strongly diminish the
ancestor subspecies, but not the other host subspecies. If possible, it would be
good to further elucidate that by sequencing the competition experiments; also with
respect to the γ parasite. If additional sequencing experiments are not possible,
can you provide other explanations?

6) While more clarity in these processes could be obtained by more competition
experiments between α parasites and hosts at different timepoints, but better
representation of the data should help.

7) The evolutionary arms race experiments of Figure 3 are a nice demonstration that
there are "alternating" fitness improvements in subsequent host/parasites
when challenged with a prior parasite/host partner. However, in light of the
complete sequencing record outlined in Figure 2, one would want to know and see more
specifically how distinct parasite/host lineages arose over the course of the
coevolution experiment, especially since this seems to be a beautiful advantage of
this RNA ecosystem. For example, can one draw a lineage map for parasite lineages
over time based on the specific rounds that the authors focused on for RNA
sequencing (13, 24, 33, 39,.…99, 104, 110)? It would seem that clusters with common
mutations could be derived at each of these snapshots. We would like to see
phylogenetic analyses of sequences as a function of time (especially at particularly
key time points of population transition, e.g., between rounds 99 and 115).

You highlight 3 dominant groups of parasites, α, β, and γ, but what can you say about
finer-grained lineage clustering and "transitions" that occur within these
3 dominant groups (e.g., parasite-α13 and parasite-α24 seem to be different
transitional forms (Figure 2A)-what exactly distinguishes these sub-strains at the
genetic level? Instead of the mutation index tables in Supplementary Figure 4 and
Figure 2—figure supplement 43, please report how strains are delineated by sets of
mutations (instead of focusing on summarizing each individual mutation and which
strains had them). The Hamming distance metric/analysis by itself is not very
satisfying for displaying/characterizing strain clusters or distinguishing genotype
centroids in Figure 2; including time/round information may help resolve
lineages.

8) What specific reciprocal mutational changes in host and parasite occur over the
course of the evolution experiment? Figure 3 demonstrates clear fitness benefit
changes of host and parasite, but it would be good to highlight the underlying
adaptive genetic changes responsible for the evolutionary arms race. This could
provide fertile ground for follow-on mechanistic studies for how host or parasite
fitness improves in response to the other.

9) You suggest, as alluded to in the title, that coevolution drives diversification
(Discussion, first paragraph), but it was not clear what you mean by this. There was
little discussion of strain or mutational diversity: what do the authors precisely
mean by "diversity" as it relates to the results of the present study. I
imagine one would need at least one (if not several) metrics of diversity, and apply
it either to the diversity of lineages in the population and/or the
distribution/spectra of mutations that are accrued in host or parasite (or
ecosystem) as a function of time. The emergence of diversity/diversification is
suggested in the title, but this theme does not seem to be adequately addressed in
the discussion of results. In the third paragraph of the Discussion, it is pointed
out that antagonistic host-parasite coevolution could increase the complexity of
individuals, but this idea also does not seem to be addressed in this study. The
results from this work seem to recapitulate a well-known and accepted idea that
parasites undergo genome reduction, but I feel the authors need to discuss more of
the implications of their findings in relation to the literature on gene loss/genome
reduction (e.g., Wolf and Koonin, 2013 and Albalat and Cañestro, 2016. What new
insight(s) on this topic is revealed by the authors' new results? I found it
intriguing that after drastic genome reduction early on, late parasite lineages had
genome expansion, which goes counter to a naïve view that the genomes of parasites
"just get smaller"-perhaps the authors' results tell us something deeper
about the conditions for genome reduction in parasites?

[Editors' note: further revisions were suggested prior to acceptance, as described
below.]

Thank you for submitting your article "Emergence and diversification of a
host-parasite RNA ecosystem through Darwinian evolution" for consideration by
*eLife*. Your article has been reviewed by three peer reviewers,
and the evaluation has been overseen by Detlef Weigel as Reviewing and Senior
Editor. The following individuals involved in review of your submission have agreed
to reveal their identity: Eörs Szathmáry (Reviewer #1); Erik Hom (Reviewer #3).

As you will see, the reviewers greatly appreciated your efforts to revise the work,
and based on their advice, I am happy in principle to accept the submission for
publication. I would like, however, to ask you to accommodate the suggestions by
reviewer #2, and consider more fully meeting the specific concern of reviewer #3,
who said that while Figure 2—figure supplement 5 is appreciated, this analysis does
not quite reveal how the different parasite lineages arise or are related to one
another in time. Please consider adding a joint temporal-sequence correlation
analysis beyond a single phylogenetic tree of lineages (Figure 2—figure supplement
5).

*Reviewer #1:*

The paper has been duly revised and it will be a valuable contribution to the field
of experimental molecular coevolution.

*Reviewer #2:*

I like to congratulate the authors on their improved manuscript, which is a really
nice and valuable addition to our knowledge on RNA evolution.

I just have 2 suggestions:

1) Rooting of phylogenetic tree. In this case a "real" root is known: the
initial host sequence. I would suggest to reroot the tree accordingly (and indeed
add the initial host)

2) In the discussion about complexity generation (Discussion) the wording suggest
that the parasite evolved more complexity by adding, whereas, as is now very clear
from the rest of the manuscript is arose as close mutant of the current dominant
host, by deletion, but retaining mor of the host genome. Please make that clear also
in this discussion, also emphasizing its co-occurence with the α parasite
lineage.

Finally, I missed what the mutation index means in the mutation table, which, by the
way is very nicely improved.

*Reviewer #3:*

This manuscript is much improved. I am satisfied with most of the responses and the
changes that the authors' made in addressing the reviewer comments. I feel the key
points of novelty and insight for the current work are much better presented,
including incorporating nuances raised in responses to Essential revision comments
#5 and #9, making this an exciting contribution.

The response to Essential revision comment #7, however, remains somewhat wanting in
my opinion. The authors did improve their argument for "reciprocal
host-parasite mutational dynamics underlying armed-race history" through
refinements in text and Figure 2—figure supplement 2, and the inclusion of the
phylogenetic tree for top 3 host and parasite lineages in Figure 2—figure supplement
5. However, it still feels like there is a missed opportunity to really describe the
co-evolutionary dynamics of this system in (1) time, and (2) that the takes into
account the specific sequence relationships between the different host and parasite
quasi-species. The uniqueness of this RNA system and the described study is that
(nearly) all the lineages of host and parasite have been sequenced over the course
of coevolution. Thus, it would seem like a lineage map (in time) could be
constructed (even if there were only snapshots of specific rounds from the
experiment); but it doesn't seem like the sequence data time series is being mined
for the fullest insights. While Figure 2—figure supplement 5 is an improvement in
showing the phylogenetic relatedness of parasite vs. host lineages, this analysis
doesn't really reveal how the different parasite lineages arise or are related to
one another in time (E.g., how does parasite-β or parasite-γ actually arise in
response to the sequences (both host and parasite) that exist leading up to their
emergence in the population (e.g., in round 104)?) Although I've not spent enough
time thinking about how best to do such an analysis, I imagine it might involve some
joint temporal-sequence correlation analysis beyond a single phylogenetic tree of
lineages (Figure 2—figure supplement 5). The color-coding of Figure 2—figure
supplement 2 is definitely an improvement from the previous version of the
manuscript and does capture some of this temporal-sequence information-but I still
find the figure hard to parse towards understanding how mutations in different
parasite size classes (α/β/γ) relate to one another and hosts over the course of
coevolution. The second paragraph of the Discussion does a good job explaining an
example of apparent reciprocal sequence changes between parasite-α and host, but one
cannot distil this from the main figures (Figure 2A shows oscillations in
concentration, but this is unrelated to reciprocal mutation changes) and it is quite
cumbersome to distil this from Figure 2—figure supplement 2. Figure 2—figure
supplement 2 is a sensible mapping of mutations to gene body sequence position but
this is only one view of the data: e.g., grouping this data according to parasite
class (α/β/γ) (or even similarity of persistence profile over time) might be
alternatively illuminating to get at or synthesize what I feel is missing in the
analysis presented (as mentioned above). The Hamming distance plots as a function of
time in Figure 2—figure supplement 4 get at this, but might be better presented as
an animation rather than a frame series; it also only focuses on a subset of host
and parasite-α sequences. I recommend the authors work to reduce salient points from
Figure 2—figure supplement 2 into an additional main figure that highlights the
reciprocal sequence changes that occur and supports the claims of reciprocal
coevolution broadly, not focused merely on parasite-α but showing the relevant
coevolutionary dynamic of all parasite size classes.

The other main suggested edit is to include a discussion in the Sequence Analysis
section in either the Materials and methods or Results for how sequence errors were
handled in relation to singleton reads. In the Supplementary file 1 (which is very
helpful), many reads are listed with a genotype frequency of 1. In the Materials and
methods section (subsection “Sequence analysis”), it is stated that circular
consensus sequencing for at least 5-10 reads was performed. It is not clear how
these two things relate, nor how the singleton sequences of Supplementary file 1
were incorporated into the main analyses described: were singletons included in all
analyses? If so, should one really include them? What thresholds were applied to
reject sequences that might be erroneous? Clarification of these points in the
manuscript would be very helpful.

---

## [Author Response]

Essential revisions:1) The idea of an arms race in this system is attractive, but it raises questions
about the role of homologous copy-choice recombination in the host RNA
population. Under such conditions of directional selection (exerted by the
parasites) recombination should be advantageous. Is there a signal of this
process in the sequences of the host population?

We agreed that the recombination among the host RNA population can be advantageous
and very interesting. We examined this possibility by using a recombination
detection program RDP4 (Martin et al., 2015). We extracted top-50 most frequent host
and also parasite sequences for every sequenced round from the aligned cDNA data set
(we provide this data upon the request from reviewer #2) and exerted exploratory
recombination scan with all the available algorithms, but unfortunately, no
recombination signal was detected. In fact, as we report in Figure 3—source data 1,
two diverged host lineages (Host-99 and Host-115) share only a few dominant
mutations that accumulated before the divergence and then underwent very distinct
evolutionary pathways without any “jumping in” of shared mutations into each other.
Therefore, while further long-term evolution might lead to beneficial recombination
events in the future, we have no clear evidence that it has happened so far. We
briefly mention that we did not detect a signal of recombination in the second
paragraph of Discussion as follows.

“Finally, we would like to mention that we searched for possible recombination events
in the host and parasite sequences throughout the replication experiment, using the
RDP4 program (Martin et al., 2015), but did not detect a recombination signal.”

2) It is confusing that different measurements of density for host (replicase) as
well as the different parasites are used. Please explain.

We are sorry for this confusion. In Figure 2A, we measured the host and parasitic
RNAs by different methods, quantitative PCR after reverse transcription (RT-qPCR)
and by quantification of band intensity after gel electrophoresis, respectively.
That is because RT-qPCR was not applicable to the parasitic RNA population, the
mixture of deletion mutants sharing the highly similar sequences as those of host
RNAs. Therefore, we could not design a primer set for RT-qPCR that specifically
detects the parasitic RNAs that appeared during evolution. In Figure 3A, on the
other hand, we knew the specific sequence of parasite RNA “clones”, thus we were
able to design primers including unique mutations that are not shared among
different RNA species. To clarify this point, we changed manuscripts in Results
section as following:

In the subsection “Population dynamics of host and parasitic RNAs”

“The parasitic RNA was measured using the band intensity after polyacrylamide gel
electrophoresis (Figure 2—figure supplement 1) because these parasites were deletion
mutants of the host RNA and could not be uniquely targeted by RT-qPCR. In some
rounds (7-12, 16-22, and 75-84), the parasitic RNAs were under the detection limit
(less than ~30 nM) and not visible due to the lower sensitivity of gel analysis
compared to that of RT-qPCR.”

In the subsection “Competitive replication assay of host and parasitic RNAs”

“RT-qPCR of parasites was possible in this experiment because we designed primers
very specific to each parasite clone, which was not possible for the evolving RNA
mixture (Figure 2A) containing various mutations.”

3) The population dynamics first shows strong oscillation, which later dampen
till t~75. Competition is only shown for parasite t-13 and host t=0 and parasite
t=13 and host t=99 (after one more deep oscillation). What happens in between?
How can the replicase sustain such high densities of host and parasite RNA,
while that is initially not the case?

In the initial stage (t = 0 – 25), the host RNA was not able to replicate in the same
compartment with the parasite-α and thus a clear oscillation pattern appeared, as
reported in our previous study (Bansho et al., 2016). In the paper, we also showed
that the host RNA at t=43 acquired partial resistance against parasite-α, which
probably allowed the co-replication of hosts and parasites at higher concentrations
exhibiting clear contrast to the initial clear oscillation. We clarified this point
by adding the following statements in the Results section (in the subsection
“Population dynamics of host and parasitic RNAs”).

“The elevation of the host RNA concentration can be attributed to less replication
inhibition by parasite-α. […] The prevalence of these mutations probably allows the
host RNA population to maintain higher concentrations than that in the early
stage.”

We further discussed population dynamics between t=13~99 in the new second paragraph
in the Discussion section (see also Essential revision comment #6).

“High-resolution sequence tracking of RNA populations allowed the observation of
reciprocal host-parasite mutational dynamics underlying evolutionary arms-race
history. […] Upon the invasion of parasite-γ in round 115, some mutations (e.g.,
C72U, C259U, U501C, and A851G) appeared and disappeared in the host population.
These host-parasite mutational dynamics exhibit how coevolution had progressed
during the replication experiment.”

4) Is the α-parasite after t=90 offspring of the older α parasite or a newly
evolved from the host by a large deletion?

Based on the sequence similarity, the parasite-α after t=90 is the mixture of
offsprings of the older ones at t=13 and deletion mutants of the evolved hosts in
later rounds. We added the explanation of this point in the Results section (in the
subsection “Sequence analysis”):

“We also observed the appearance of new parasite-α species from the evolved host RNAs
owing to deletion. […] We could not obtain the sequence data of para”.

5) The β and γ parasites are clearly offspring of the later hosts, each of one of
the evolved 'subspecies' of the host. The β parasites appear to strongly
diminish the ancestor subspecies, but not the other host subspecies. If
possible, it would be good to further elucidate that by sequencing the
competition experiments; also with respect to the γ parasite. If additional
sequencing experiments are not possible, can you provide other explanations?

If our understanding is correct, the reviewer is asking whether parasite-β or -γ
specifically diminish each ancestor host RNA, Host-99 or Host-115, respectively.
That is an interesting question and such specificity would be reasonable if each of
the parasites appeared from each of the host RNA through adaptive evolution. To
answer this question, competition experiments all four combinations of Hosts-99 and
-115 with parasite-β and -γ. Since three of the experiments (Host-99 vs.
parasite-β99, Host-115 vs. parasite-β99, and Host-115 vs. parasite-γ115) has been
already performed (Figure 3), we newly conducted the remaining competition
experiment (Host-99 vs. Parasite-γ115) and confirmed that Parasite-γ115 is hardly
replicated by Host-99 (new Figure 3—figure supplement 1). This result indicates that
the parasite-β and -γ selectively “infect” each of their ancestors, Host-99 and
Host-115, respectively. We added the explanation of this result in the Results
section as follows.

“We also examined the Host-99 vs. Parasite-γ115 relationship and found that
Parasite-γ115 was hardly replicated by Host-99 (Figure 3—figure supplement 1),
indicating that parasite-β and parasite-γ are specifically parasitic to Host-99 and
Host-115, respectively.”

6) While more clarity in these processes could be obtained by more competition
experiments between α parasites and hosts at different timepoints, but better
representation of the data should help.

We are thankful for the useful suggestion. We agreed that the evolution of parasite-α
is another interesting theme, and thus we would like to address it in the next
study. Instead of the additional experiments, we added a new paragraph in the
Discussion section as shown in the response to Essential revision comment #3 above
to describe host-parasite evolutionary processes in detail, including the arms race
between the host and the parasite-α (indicated with underlines).

7) The evolutionary arms race experiments of Figure 3 are a nice demonstration
that there are "alternating" fitness improvements in subsequent
host/parasites when challenged with a prior parasite/host partner. However, in
light of the complete sequencing record outlined in Figure 2, one would want to
know and see more specifically how distinct parasite/host lineages arose over
the course of the coevolution experiment, especially since this seems to be a
beautiful advantage of this RNA ecosystem. For example, can one draw a lineage
map for parasite lineages over time based on the specific rounds that the
authors focused on for RNA sequencing (13, 24, 33, 39,.…99, 104, 110)? It would
seem that clusters with common mutations could be derived at each of these
snapshots. We would like to see phylogenetic analyses of sequences as a function
of time (especially at particularly key time points of population transition,
e.g., between rounds 99 and 115).You highlight 3 dominant groups of parasites, α, β, and γ, but what can you say
about finer-grained lineage clustering and "transitions" that occur
within these 3 dominant groups (e.g., parasite-α13 and parasite-α24 seem to be
different transitional forms (Figure 2A)-what exactly distinguishes these
sub-strains at the genetic level? Instead of the mutation index tables in
Supplementary Figure 4 and Figure 3, please report how strains are delineated by
sets of mutations (instead of focusing on summarizing each individual mutation
and which strains had them). The Hamming distance metric/analysis by itself is
not very satisfying for displaying/characterizing strain clusters or
distinguishing genotype centroids in Figure 2; including time/round information
may help resolve lineages.

Upon this comment, we newly added phylogenic analysis of the top 3 most frequent of
host and parasite sequences in all the sequenced rounds (Figure 2—figure supplement
5). The explanation of this analysis is included in the Results section in the
revised manuscript as follows.

“To understand the relationship between the host and parasite lineages, we performed
phylogenic analysis of the top 3 most frequent genotypes of the host and parasite
RNAs in all the sequenced rounds (Figure 2—figure supplement 5). […] It is also
notable that some parasite-α genotypes are located within host clusters (Α 072R
Rank2, Α 094R Rank2, and Α 099 Rank1 in Figure 2—figure supplement 5 with green
ticks), indicating that they emerged from the evolved hosts in later rounds.”

We also improved Figure 2—figure supplement 3 with heatmaps of mutation fixation
dynamics for all the RNA species to easily grasp transitions of major sequences and
existing mutations.

8) What specific reciprocal mutational changes in host and parasite occur over
the course of the evolution experiment? Figure 3 demonstrates clear fitness
benefit changes of host and parasite, but it would be good to highlight the
underlying adaptive genetic changes responsible for the evolutionary arms race.
This could provide fertile ground for follow-on mechanistic studies for how host
or parasite fitness improves in response to the other.

We detailed how host-parasite coevolution has progressed with reciprocal mutational
changes in a newly added paragraph in Discussion (please see response to Essential
revisions comment #3).

9) You suggest, as alluded to in the title, that coevolution drives
diversification (Discussion, first paragraph), but it was not clear what you
mean by this. There was little discussion of strain or mutational diversity:
what do the authors precisely mean by "diversity" as it relates to the
results of the present study. I imagine one would need at least one (if not
several) metrics of diversity, and apply it either to the diversity of lineages
in the population and/or the distribution/spectra of mutations that are accrued
in host or parasite (or ecosystem) as a function of time. The emergence of
diversity/diversification is suggested in the title, but this theme does not
seem to be adequately addressed in the discussion of results.

As you and reviewer #3 point out, we agree that we should more clearly state what is
diversity in the Discussion section. By the word “diversity”, we mean that there are
two or three distinct lineages (or branches) in both host and parasite RNAs. To
clarify this point, we changed a sentence in the Abstract and also largely modified
the first paragraph of the Discussion section as shown below.

Abstract

“In prebiotic evolution, molecular self-replicators are considered to develop into
diverse, complex living organisms. […] These results provide the first experimental
evidence that the coevolutionary interplay between host-parasite molecules plays a
key role in generating diversity and complexity in prebiotic molecular
evolution.”

Discussion section

“Coevolution of host and parasitic replicators is a major driver in the evolution of
life. In this study, we investigated the Darwinian evolution process of an RNA
replication system and demonstrated the emergence of a host-parasite ecosystem in
which new types of host and parasitic RNAs appeared successively and exhibited
antagonistic coevolutionary dynamics. […] Therefore, evolutionary arms races between
host-parasite molecules may have been an important mechanism to generate and
maintain diversity in molecular ecosystems before the origin of life.”

In the third paragraph of the Discussion, it is pointed out that antagonistic
host-parasite coevolution could increase the complexity of individuals, but this
idea also does not seem to be addressed in this study. The results from this
work seem to recapitulate a well-known and accepted idea that parasites undergo
genome reduction, but I feel the authors need to discuss more of the
implications of their findings in relation to the literature on gene loss/genome
reduction (e.g., Wolf and Koonin, 2013 and Albalat and Cañestro, 2016. What new
insight(s) on this topic is revealed by the authors' new results? I found it
intriguing that after drastic genome reduction early on, late parasite lineages
had genome expansion, which goes counter to a naïve view that the genomes of
parasites "just get smaller" – perhaps the authors' results tell us
something deeper about the conditions for genome reduction in parasites?

We thank the reviewer for the important suggestion. Regarding the complexity of
individuals, as the reviewer pointed out, our message was that late parasites (-β
and -γ) had expanded genome compared to the early parasite (-α), which may have been
caused by coevolution. This genome expansion of parasites has never been observed at
least in Qβ RNA-based molecular evolution studies (there are not many other
artificial replication systems available yet) and also seems to be against the trend
of reductive evolution observed in nature (Wolf and Koonin, 2013, Morris, Lenski and
Zinser, 2012). To clarify the implication of our results in a reductive evolution
context, we largely modified the fourth paragraph of the Discussion section as
follows.

“It is generally believed that evolution progresses toward more complexity in nature
(Petrov, 2001; Sharov, 2006); however, genome reduction is also a popular mode of
evolution (Albalat and Cañestro, 2016; Morris et al., 2012; Wolf and Koonin, 2013).
[…] The next important question would be whether further long-term coevolution can
lead to genome expansion of the host RNA.”

[Editors' note: further revisions were suggested prior to acceptance, as described
below.]

As you will see, the reviewers greatly appreciated your efforts to revise the
work, and based on their advice, I am happy in principle to accept the
submission for publication. I would like, however, to ask you to accommodate the
suggestions by reviewer #2, and consider more fully meeting the specific concern
of reviewer #3, who said that while Figure 2—figure supplement 5 is appreciated,
this analysis does not quite reveal how the different parasite lineages arise or
are related to one another in time. Please consider adding a joint
temporal-sequence correlation analysis beyond a single phylogenetic tree of
lineages (Figure 2—figure supplement 5).

We are grateful to the reviewers for their effort to read and comments on our
manuscript, which improved our manuscript significantly. In accordance with the
reviewers’ advice, we have changed some figures and revised our manuscript. We
believe that the reviewers’ comments have now been addressed in the revision.

Reviewer #2:I like to congratulate the authors on their improved manuscript, which is a
really nice and valuable addition to our knowledge on RNA evolution.I just have 2 suggestions:1) Rooting of phylogenetic tree. In this case a "real" root is known:
the initial host sequence. I would suggest to reroot the tree accordingly (and
indeed add the initial host)

Following the reviewer’s suggestion, we added the initial host (Ancester host,
indicated with the blue asterisk) in the phylogenic three (Figure 2—figure
supplement 4).

2) In the discussion about complexity generation (Discussion) the wording suggest
that the parasite evolved more complexity by adding, whereas, as is now very
clear from the rest of the manuscript is arose as close mutant of the current
dominant host, by deletion, but retaining mor of the host genome. Please make
that clear also in this discussion, also emphasizing its co-occurence with the α
parasite lineage.

We agreed that the previous description was misleading. We rewrote the corresponding
part as follows to clarify that the new parasites appeared from the evolved host
through deletion.

“The new parasites, parasite-β and parasite-γ, became longer because they retained a
part of the M-site sequence, a recognition site for Qβ replicase (Meyer et al.,
1981; Schuppli et al., 1998), which did not exist in parasite-α. […] According to
this scenario, the new parasites appear to be expanding the genomic information to
cope with the evolved strategy of the host RNA, which may be consistent with recent
theoretical studies that suggest that host-parasite antagonistic coevolution is an
effective mechanism to increase the complexity of individuals (Seoane and Solé,
2019; Zaman et al., 2014).”

Reviewer #3:This manuscript is much improved. I am satisfied with most of the responses and
the changes that the authors' made in addressing the reviewer comments. I feel
the key points of novelty and insight for the current work are much better
presented, including incorporating nuances raised in responses to Essential
revision comments #5 and #9, making this an exciting contribution.The response to Essential revision comment #7, however, remains somewhat wanting
in my opinion. […] I recommend the authors work to reduce salient points from
Figure 2—figure supplement 2 into an additional main figure that highlights the
reciprocal sequence changes that occur and supports the claims of reciprocal
coevolution broadly, not focused merely on parasite-α but showing the relevant
coevolutionary dynamic of all parasite size classes.

We agreed that the reciprocal evolutionary progress of the host and the parasites was
difficult to grasp from our previous figures. According to the reviewer’s
recommendation, we refined Figure 2—figure supplement 4 (Round-by-round 2D mapping
of top genotypes) to include parasites-β and -γ and moved to a main figure (new
Figure 3). We also included an animation of the same data (new Figure 3—figure
supplement 1). We hope that these figures help readers to grasp the co-evolutionary
dynamics.

The other main suggested edit is to include a discussion in the Sequence Analysis
section in either the Materials and methods or Results for how sequence errors
were handled in relation to singleton reads. In the Supplementary file 1 (which
is very helpful), many reads are listed with a genotype frequency of 1. In the
Materials and methods section (subsection “Sequence analysis”), it is stated
that circular consensus sequencing for at least 5-10 reads was performed. It is
not clear how these two things relate, nor how the singleton sequences of
Supplementary file 1 were incorporated into the main analyses described: were
singletons included in all analyses? If so, should one really include them? What
thresholds were applied to reject sequences that might be erroneous?
Clarification of these points in the manuscript would be very helpful.

In this study, all sequences were read more than 5 times to minimize the effect of
sequence error and the resultant circular consensus sequences (CCS) were used for
analyses. The read number in the Supplementary file 1 is those of the CCS that are
obtained from at least 5-10 reads. This point was not clearly written in the
previous manuscript. Then, we rewrote the corresponding parts and added a short
explanation about the method to minimize sequencing errors in the Materials and
methods section.

“Sequence analysis

The RNA mixtures of rounds 13, 24, 33, 39, 43, 50, 53, 60, 65, 72, 86, 91, 94, 99,
104, 110, and 115 in the long-term replication experiment were purified with spin
columns (RNeasy, QIAGEN). […] In the subsequent analysis, we focused on only the
genotypes associated with these 72 mutation sites. Focusing only on these dominant
mutation sites minimizes the influence of remaining sequencing errors and
non-dominant mutations in the other sites.”